# Q-FSRU: Quantum-Augmented Frequency-Spectral Fusion for Medical Visual Question Answering

## Abstract

Solving tough clinical questions that require both image and text understanding is still a major challenge in healthcare AI. In this work, we propose Q-FSRU, a new model that combines Frequency Spectrum Representation and Fusion (FSRU) with a method called Quantum Retrieval-Augmented Generation (Quantum RAG) for medical Visual Question Answering (VQA). The model takes in features from medical images and related text, then shifts them into the frequency domain using Fast Fourier Transform (FFT). This helps it focus on more meaningful data and filter out noise or less useful information. To improve accuracy and ensure that answers are based on real knowledge, we add a quantum inspired retrieval system. It fetches useful medical facts from external sources using quantum-based similarity techniques. These details are then merged with the frequency-based features for stronger reasoning. We evaluated our model using the VQA-RAD dataset, which includes real radiology images and questions. The results showed that Q-FSRU outperforms earlier models, especially on complex cases needing image text reasoning. The mix of frequency and quantum information improves both performance and explainability. Overall, this approach offers a promising way to build smart, clear, and helpful AI tools for doctors.

## 1 Introduction

Medical visual question answering (Med-VQA) represents an emerging interdisciplinary challenge that sits at the intersection of computer vision, natural language processing, and clinical decision-making (Lin et al., 2023). In real-world clinical environments, radiologists and medical practitioners frequently interact with imaging studies by formulating diagnostic questions such as 'Is there evidence of a pulmonary nodule?' or 'Does this MRI show signs of cerebral edema?'. Addressing such queries demands not only sophisticated understanding of visual content in medical images but also deep contextual knowledge and nuanced language comprehension (Lau et al., 2018). The development of AI systems for Med-VQA faces several unique challenges that distinguish it from general-domain VQA. These include severe data scarcity due to privacy concerns, highly specialized medical terminology, complex imaging modalities (CT, MRI, X-ray, etc.), and the critical nature of medical decision-making where errors can have serious consequences. While transformer-based architectures and cross-modal fusion techniques have shown remarkable progress in general VQA benchmarks (Antol et al., 2015; Vaswani et al., 2017), their direct application to medical domains has yielded limited success. Recent medical-specific vision-language models such as LLaVA-Med (Li et al., 2023), STLLaVA-Med (Sun et al., 2024a), and concept-aligned approaches like MMCAP (Yan et al., 2024) have improved domain adaptation, but they predominantly operate in the spatial domain, potentially overlooking subtle frequency-based patterns that are particularly relevant in medical imaging. Most current Med-VQA models rely on convolutional or attention-based feature extractors that process images in the spatial domain. While effective for capturing local structures, these approaches may miss global contextual cues embedded in frequency spectra that are especially important for detecting pathological patterns in medical images (Cai et al., 2023). Concurrently, retrieval-augmented methods that incorporate external knowledge have shown promise in improving factual grounding (Lewis et al., 2021), but they typically rely on classical similarity measures like cosine similarity, which may not fully capture the complex semantic relationships required for

clinical reasoning. Recent work has demonstrated the effectiveness of frequency-domain representations in various multimodal tasks. As shown by Lao et al. (2024), frequency spectrum analysis can be more effective for multimodal representation and fusion in rumor detection, while Cai et al. (2023) proposed FDTrans, a frequency-domain transformer for multimodal medical image analysis. In medical imaging specifically, frequency-aware components have been incorporated into architectures like FreqU-FNet (Xing, 2025) for segmentation tasks. However, these approaches have not been comprehensively explored for medical VQA, where the combination of visual and textual frequency analysis could potentially capture complementary diagnostic information. To address these limitations, we propose Q-FSRU, a novel framework that combines Frequency Spectrum Representation and Fusion (FSRU) with a Quantum-inspired Retrieval-Augmented Generation (Quantum RAG) mechanism for medical VQA. Our approach is motivated by two key insights: first, that transforming multimodal features into the frequency domain can help capture global contextual patterns often missed by spatial processing; and second, that quantum-inspired similarity measures may offer advantages over classical retrieval methods for capturing nuanced semantic relationships in medical knowledge. The frequency fusion component of Q-FSRU transforms input features from both image and text modalities using Fast Fourier Transform (FFT), allowing the model to selectively attend to salient frequency-domain signals while suppressing irrelevant spatial noise. This spectral transformation enables our model to capture global contextual cues that are particularly valuable for identifying pathological patterns in medical images. To complement this, we integrate a quantum-inspired retrieval mechanism that fetches relevant external clinical knowledge based on amplitude-based similarity principles, helping ground the model's reasoning in verifiable medical facts. Our contributions can be summarized as follows:

1. We introduce a novel frequency domain fusion framework for medical VQA that transforms visual and textual features using FFT to capture complementary spectral patterns.

2. We propose a quantum-inspired retrieval mechanism that enhances factual grounding by retrieving relevant medical knowledge based on amplitude similarity measures.

3. We demonstrate through extensive experiments on the VQA-RAD dataset that our approach achieves competitive performance compared to existing methods, with particular strengths in complex reasoning cases.

4. We provide analysis showing that the combination of spectral processing and knowledge retrieval improves both performance and interpretability, making the model more suitable for clinical applications.

## 2 RELATED WORK

### 2.1 MEDICAL VISUAL QUESTION ANSWERING

Medical Visual Question Answering (Med-VQA) is a core challenge in healthcare AI, requiring joint reasoning over medical images and domain-specific language. Early efforts adapted general VQA frameworks to clinical data but struggled with specialized terminology and imaging complexity (Lau et al., 2018; Lin et al., 2023). More recent approaches such as STLLaVA-Med (Sun et al., 2024b) leverage large language models and self-training strategies, achieving notable gains through domain adaptation. However, most existing methods operate solely in the spatial domain and have limited ability to capture frequency-based patterns that may hold diagnostic value. Furthermore, knowledge integration remains constrained by conventional retrieval techniques. To address these gaps, we propose a framework that combines frequency-domain representations with quantum-inspired retrieval to better align image-text reasoning with clinical requirements.

### 2.2 FREQUENCY-DOMAIN REPRESENTATIONS

Frequency-domain analysis has demonstrated value across various computer vision applications. In medical imaging specifically, Cai et al. (2023) developed FDTrans, a frequency-domain transformer that captures complementary information to spatial representations for diagnostic tasks. This work highlights how spectral features can enhance medical image analysis beyond conventional approaches. Xing (2025) incorporated frequency-aware components into segmentation architectures, showing improved performance on imbalanced medical datasets through better global pattern

capture. The work by Lao et al. (2024) is particularly relevant, showing that frequency spectrum analysis improves multimodal representation and fusion for rumor detection. However, their focus on social media content differs from our medical application, and they did not explore knowledge retrieval mechanisms. Our approach extends this foundation by applying frequency-domain methods specifically to medical visual question answering while incorporating novel retrieval components.

## 2.3 Quantum-Inspired Methods in Information Retrieval

Quantum-inspired approaches to information retrieval have developed over the past two decades, offering alternative mathematical frameworks for similarity measurement and representation learning. As surveyed by Uprety et al. (2021), quantum theory provides a generalized probability and logic framework that has shown promise for developing more dynamic and context-aware retrieval systems. This established research area offers theoretical foundations for our quantum-inspired retrieval approach. Recent applications demonstrate the practical value of quantum-inspired methods. Kankeu et al. (2025) proposed quantum-inspired projection heads and similarity metrics for representation learning, showing competitive performance with significantly reduced parameters compared to classical methods. Their work on embedding compression for information retrieval tasks provides direct precedent for our quantum-inspired similarity approach. In computer vision applications, Nguyen et al. (2025) developed Quantum-Brain, a quantum-inspired neural network for vision-brain understanding problems. Their approach demonstrates how quantum principles can enhance connectivity learning in neural representations, particularly relevant for tasks requiring complex relationship modeling. This work shows the applicability of quantum-inspired methods to vision-related tasks similar to medical visual question answering. These quantum-inspired approaches differ from our work in their specific applications, but collectively establish the viability of quantum principles for enhancing similarity measurement and representation learning. Our contribution lies in adapting these principles specifically for medical knowledge retrieval in visual question answering contexts.

## 2.4 Knowledge Retrieval in Visual Question Answering

Retrieval-augmented methods have become increasingly important for tasks requiring external knowledge integration. The foundational work by Lewis et al. (2021) established retrieval-augmented generation as a powerful approach for knowledge-intensive tasks. In medical contexts, however, standard retrieval methods often struggle with the nuanced relationships required for clinical reasoning. Recent multimodal research continues to advance integration techniques. Huang et al. (2025) explored pixel-level insight for biomedical applications, while datasets like MMVP from Zhang et al. (2024) provide resources for evaluating multimodal systems. These contributions highlight the ongoing importance of robust multimodal integration in healthcare applications.

### 2.4.1 Research Contributions

Our work distinguishes itself from existing approaches through several key contributions. While prior frequency-domain methods that focus on single modalities or non-medical applications, we specifically address medical visual question answering with integrated frequency processing. Compared to standard retrieval approaches, we introduce quantum-inspired similarity measures grounded in established research. And unlike conventional medical visual question answering systems, we combine both frequency-domain analysis and quantum-inspired retrieval within a unified framework, Q-FSRU designed for clinical applications. The integration of these components addresses limitations in current medical visual question answering systems while building on established research in frequency-domain processing and quantum-inspired information retrieval. This combination represents a novel approach to enhancing both performance and interpretability in medical artificial intelligence systems.

## 3 Problem Definition

We formulate medical visual question answering as a multimodal classification task. Given the VQA-RAD dataset $\mathcal{D} = \{(I_i, Q_i, y_i)\}_{i=1}^N$, where $I_i \in \mathbb{R}^{H \times W \times 3}$ represents a medical image, $Q_i$ denotes a clinical question, and $y_i \in \{0, 1, \ldots, C-1\}$ indicates the answer class among $C$ possible

categories. The VQA-RAD dataset contains both binary ("yes"/"no") and open-ended questions; we focus on the subset with categorical answers suitable for classification, filtering questions to those with discrete answer classes. The objective is to learn a mapping function $f : (I_i, Q_i) \rightarrow \hat{y}_i$ that predicts the correct answer. Our Q-FSRU model enhances this mapping through two key components:

- **Frequency-spectral fusion**: $z_i^{\text{freq}} = f_{\text{FSRU}}(I_i, Q_i)$ transforms multimodal features into the frequency domain
- **Knowledge retrieval**: $k_i \in \mathbb{R}^d$ represents relevant medical knowledge retrieved from external corpora
- **Feature integration**: $\hat{y}_i = f_\theta(z_i^{\text{freq}}, k_i) = \text{MLP}([z_i^{\text{freq}} \| k_i])$ where $\|$ denotes concatenation

The model is trained to minimize a combined objective function:

$$\mathcal{L} = \mathcal{L}_{\text{CE}}(\hat{y}_i, y_i) + \alpha \mathcal{L}_{\text{intra}} + \beta \mathcal{L}_{\text{cross}}$$

where $\mathcal{L}_{\text{CE}}$ represents the cross-entropy classification loss, $\mathcal{L}_{\text{intra}}$ and $\mathcal{L}_{\text{cross}}$ denote intra-modal and cross-modal contrastive losses respectively, and $\alpha$, $\beta$ are hyperparameters that balance the contrastive objectives. leverages frequency-domain patterns and external medical knowledge while preserving the discriminative power needed for accurate clinical question answering.

# 4 METHODOLOGY

## 4.1 MODEL ARCHITECTURE OVERVIEW

The Q-FSRU framework integrates four core components: (1) multimodal feature extraction, (2) frequency-domain processing via Fast Fourier Transform, (3) quantum-inspired knowledge retrieval, and (4) multimodal fusion with contrastive learning. The architecture processes medical images and clinical questions through a sequential pipeline where frequency-domain enhancement precedes knowledge retrieval, ensuring optimal feature representation before external knowledge integration.

## 4.2 MULTIMODAL FEATURE EXTRACTION

### 4.2.1 TEXT FEATURE ENCODING

Clinical questions are processed using a pretrained word embedding approach. Given a tokenized question $Q = [w_1, w_2, \ldots, w_L]$ of length $L$, each word $w_i$ is mapped to a 300-dimensional vector using domain-specific embeddings:

$$E_{\text{text}} = \text{Embedding}(Q) \in \mathbb{R}^{L \times 300}$$

The sequence undergoes mean pooling across the temporal dimension followed by linear projection:

$$\vec{t} = W_t \cdot \left( \frac{1}{L} \sum_{i=1}^{L} \vec{e}_i \right) + b_t \in \mathbb{R}^{d_{\text{model}}}$$

where $W_t \in \mathbb{R}^{d_{\text{model}} \times 300}$, $b_t \in \mathbb{R}^{d_{\text{model}}}$, and $d_{\text{model}} = 256$.

### 4.2.2 IMAGE FEATURE ENCODING

Medical images are processed using a Vision Transformer (ViT-B/16) backbone pretrained on ImageNet. Each image $I \in \mathbb{R}^{3 \times 224 \times 224}$ is divided into 16×16 patches and processed through 12 transformer layers:

$$v = \text{ViT-B/16}(I) \in \mathbb{R}^{768}$$

The 768-dimensional output is projected to match the model dimension:

$$v_{\text{proj}} = W_v \cdot v + b_v \in \mathbb{R}^{256}$$

where $W_v \in \mathbb{R}^{256 \times 768}$, $b_v \in \mathbb{R}^{256}$.

### 4.3 FREQUENCY SPECTRUM REPRESENTATION AND FUSION

#### 4.3.1 FAST FOURIER TRANSFORM APPLICATION

To capture global contextual patterns in both modalities, the text and image features are transformed into the frequency domain using a 1D Fast Fourier Transform (FFT) applied along the feature dimension.

Let

- $t \in \mathbb{R}^{d_{\text{model}}}$ denote the input text feature vector after token embedding and encoding,
- $v_{\text{proj}} \in \mathbb{R}^{d_{\text{model}}}$ denote the projected image feature vector obtained from the visual encoder.

The 1D FFT is applied to each feature vector to obtain complex-valued frequency representations:

$$\mathcal{F}(t), \mathcal{F}(v_{\text{proj}}) \in \mathbb{C}^{d_{\text{model}}}.$$

For computational efficiency and stability, we retain only the real-valued magnitude spectrum:

$$t_{\text{freq}} = |\mathcal{F}(t)| \in \mathbb{R}^{d_{\text{model}}}, \quad v_{\text{freq}} = |\mathcal{F}(v_{\text{proj}})| \in \mathbb{R}^{d_{\text{model}}}.$$

#### 4.3.2 UNIMODAL SPECTRUM COMPRESSION

Learnable filter banks compress the frequency representations using parameterized convolution. For each modality $m \in \{\text{text}, \text{image}\}$:

$$f_m^{(k)} = \sum_{j=1}^{d_{\text{model}}} W_{\text{filter}}^{(k,j)} \cdot m_{\text{freq}}^{(j)} + b_{\text{filter}}^{(k)}$$

where $k = 1, \ldots, 4$ indexes the filter banks, and $W_{\text{filter}} \in \mathbb{R}^{4 \times d_{\text{model}}}$ are learnable parameters.

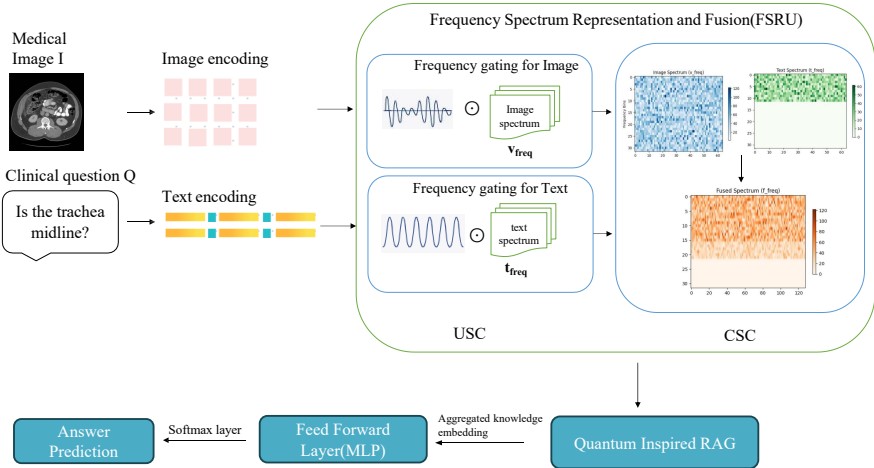

Figure 1: The architecture of the proposed Q-FSRU model for Medical Visual Question Answering. It integrates four main components: multimodal feature extraction, frequency-domain enhancement via FFT, quantum-inspired knowledge retrieval, and multimodal fusion with contrastive learning. Together, these modules enable effective reasoning over medical images and clinical questions.

#### 4.3.3 CROSS-MODAL CO-SELECTION

A gated attention mechanism enables mutual feature enhancement:

$$g_{\text{text}} = \sigma(W_{\text{gate1}} \cdot \text{AvgPool}(v_{\text{compressed}}))$$

$$t_{\text{enhanced}} = t_{\text{compressed}} \odot g_{\text{text}}$$

$$g_{\text{image}} = \sigma(W_{\text{gate2}} \cdot \text{AvgPool}(t_{\text{compressed}}))$$

$$v_{\text{enhanced}} = v_{\text{compressed}} \odot g_{\text{image}}$$

where $\sigma$ is the sigmoid function and $\odot$ denotes element-wise multiplication.

### 4.4 QUANTUM-INSPIRED RETRIEVAL AUGMENTATION

#### 4.4.1 QUANTUM STATE REPRESENTATION

Following established quantum information principles (Uprety et al., 2021; Kankeu et al., 2025), we represent features as pure quantum states. For an embedding vector $x \in R^d$, the corresponding quantum state is:

$$|\psi(x)\rangle = \frac{x}{\|x\|_2} \in \mathbb{C}^d$$

The density matrix formulation provides statistical robustness:

$$\rho(x) = |\psi(x)\rangle\langle\psi(x)| \in \mathbb{C}^{d \times d}$$

#### 4.4.2 QUANTUM FIDELITY MEASUREMENT

The similarity between query features $q$ and knowledge base entries $k_i$ is computed using the Uhlmann fidelity measure:

$$\text{Fid}(\rho_q, \rho_{k_i}) = \left(\text{Tr}\sqrt{\sqrt{\rho_q}\rho_{k_i}\sqrt{\rho_q}}\right)^2$$

This measure satisfies the quantum fidelity properties: $\text{Fid}(\rho, \rho) = 1$ and $0 \leq \text{Fid}(\rho_1, \rho_2) \leq 1$.

#### 4.4.3 KNOWLEDGE RETRIEVAL PIPELINE

The retrieval process operates after frequency processing:

1. **Query Formation**: $q_{\text{multi}} = \frac{1}{2}(t_{\text{enhanced}} + v_{\text{enhanced}})$

2. **Similarity Computation**: $\text{Sim}_i = \text{Fid}(\rho(q_{\text{multi}}), \rho(k_i))$

3. **Top-K Retrieval**: $\mathcal{K}_{\text{retrieved}} = \text{Top3}(\{\text{Sim}_i\}_{i=1}^N)$

4. **Knowledge Aggregation**: $k_{\text{agg}} = \sum_{j=1}^3 \text{softmax}(\text{Sim}_j/\tau) \cdot k_j$

where $\tau = 0.1$ is the softmax temperature.

### 4.5 MULTIMODAL FUSION AND CLASSIFICATION

#### 4.5.1 FEATURE INTEGRATION PIPELINE

The model employs a sequential integration strategy:

$$
\begin{aligned}
\text{Step 1:} \quad & t_{\text{freq}}, v_{\text{freq}} = \text{FrequencyProcessing}(t, v) \\
\text{Step 2:} \quad & k_{\text{agg}} = \text{QuantumRAG}(t_{\text{freq}}, v_{\text{freq}}) \\
\text{Step 3:} \quad & z_{\text{concat}} = [t_{\text{freq}} \| v_{\text{freq}} \| k_{\text{agg}}] \in \mathbb{R}^{3d_{\text{model}}} \\
\text{Step 4:} \quad & z_{\text{final}} = \text{MLP}_{\text{classifier}}(z_{\text{concat}})
\end{aligned}
$$

This ensures frequency-enhanced features guide the knowledge retrieval process.

### 4.5.2 MULTI-LAYER PERCEPTRON CLASSIFIER

The classification head employs a three-layer MLP with progressive dimensionality reduction. The fused input consists of only the frequency-enhanced text and image features concatenated, excluding the quantum knowledge embeddings:

$$
\begin{aligned}
z_{\text{concat}} &= [t_{\text{freq}} \| v_{\text{freq}}] \in \mathbb{R}^{2d_{\text{model}}} \\
h_1 &= \text{LayerNorm}(W_1 \cdot z_{\text{concat}} + b_1), \quad W_1 \in \mathbb{R}^{1024 \times 512} \\
a_1 &= \text{GELU}(h_1) \\
d_1 &= \text{Dropout}(a_1, p = 0.1) \\
h_2 &= \text{LayerNorm}(W_2 \cdot d_1 + b_2), \quad W_2 \in \mathbb{R}^{256 \times 1024} \\
a_2 &= \text{GELU}(h_2) \\
d_2 &= \text{Dropout}(a_2, p = 0.1) \\
\hat{y} &= W_3 \cdot d_2 + b_3, \quad W_3 \in R^{C \times 256}
\end{aligned}
$$

The architecture follows a $512 \rightarrow 1024 \rightarrow 256 \rightarrow C$ dimensionality progression with LayerNorm and GELU activations after each linear layer except the final classification layer. Softmax is applied externally during loss computation.

### 4.5.3 DUAL CONTRASTIVE LEARNING FRAMEWORK

The model employs a multi-scale contrastive learning approach with modality-specific temperatures:

$$
\begin{aligned}
\mathcal{L}_{\text{intra}} &= \frac{1}{2} \left( \mathcal{L}_{\text{contrastive}}(t, t_{\text{aug}}; \tau = 0.07) + \mathcal{L}_{\text{contrastive}}(v, v_{\text{aug}}; \tau = 0.07) \right) \\
\mathcal{L}_{\text{cross}} &= \mathcal{L}_{\text{contrastive}}(t, v; \tau = 0.05) \\
\mathcal{L}_{\text{contrastive}}(x, y; \tau) &= -\log \frac{\exp(\text{sim}(x, y)/\tau)}{\sum_{j=1}^{B} \exp(\text{sim}(x, y_j)/\tau)}
\end{aligned}
$$

where sim denotes cosine similarity and $B$ is the batch size.

### 4.5.4 COMPLETE OPTIMIZATION OBJECTIVE

The total training objective is computed as:

$$
\mathcal{L}_{\text{total}} = \mathcal{L}_{\text{CE}} + \left( 0.3 \cdot \frac{\mathcal{L}_{\text{intra-text}} + \mathcal{L}_{\text{intra-image}}}{2} + 0.7 \cdot \mathcal{L}_{\text{cross}} \right)
$$

where intra-modal losses use temperature $\tau = 0.07$, cross-modal loss uses $\tau = 0.05$, and the combined contrastive loss is added directly to the cross-entropy classification loss.

## 5 EXPERIMENTS

### 5.1 EXPERIMENTAL SETUP AND IMPLEMENTATION DETAILS

We conduct comprehensive evaluations on two established medical visual question answering benchmarks.

VQA-RAD Dataset: This benchmark comprises 3,515 clinically relevant question–answer pairs derived from radiology images spanning multiple imaging modalities, including X-rays, Computed Tomography (CT), and Magnetic Resonance Imaging (MRI). The dataset includes both binary (yes/no) and open-ended questions authored by medical experts.

PathVQA Dataset: To evaluate generalization capabilities beyond radiology domains, we include PathVQA, which contains 32,799 question–answer pairs from 4,998 pathology images. This dataset provides a larger-scale evaluation and tests domain adaptation performance when models are applied to different medical specialties. For cross-dataset experiments, we employ zero-shot transfer

learning, where models trained on VQA-RAD are directly evaluated on PathVQA without additional fine-tuning.

Data Preprocessing: All medical images are resized to 224×224 pixels and normalized using ImageNet statistics. Clinical questions are tokenized using a medical-domain vocabulary and truncated/padded to a maximum length of 50 tokens. We apply standard data augmentation techniques including random horizontal flipping and color jittering to improve robustness.

Implementation Details: The model was implemented in PyTorch using Adam optimization with learning rate $5 \times 10^{-5}$ and L2 regularization weight $10^{-5}$. Training employed 5-fold cross-validation with batch size 32 for 50 epochs maximum, using step-based learning rate decay (factor 0.98 every 5 epochs) and early stopping patience of 10 epochs. The frequency processor used $K = 4$ filter banks, and quantum retrieval retrieved $K = 3$ knowledge passages per query using direct similarity computation. To prevent information leakage, all questions for a given image are kept in the same fold, ensuring strict patient-level separation between training and validation/test splits.

## 6 Baseline Methods

We compare Q-FSRU with five types of existing methods: general-purpose VQA models (MCAN, LXMERT), medical-specific vision-language models (LLaVA-Med, STLLaVA-Med), knowledge-augmented methods (LaPA), frequency-domain approaches (FSRU), and ablation versions of our model. On the VQA-RAD dataset, Q-FSRU performs the best across all metrics, improving accuracy, F1-score, precision, recall, and AUC by 2.9–3.0 points compared to the strongest baseline. These improvements are statistically significant ($p$-value $< 0.01$).

## 7 Results and Analysis

### 7.1 Main Results on VQA-RAD

Table 1: Performance comparison on VQA-RAD dataset. Q-FSRU achieves statistically significant improvements across all metrics.

| Method | Accuracy | F1-Score | Precision | Recall | AUC | Params (M) |
|---|---|---|---|---|---|---|
| MCAN (Yu et al., 2019) | $78.3 \pm 1.2$ | $72.1 \pm 1.5$ | $75.8 \pm 1.3$ | $69.4 \pm 1.8$ | $0.842 \pm 0.02$ | 45.2 |
| LXMERT (Tan & Bansal, 2019) | $81.5 \pm 1.1$ | $75.3 \pm 1.4$ | $78.9 \pm 1.2$ | $72.8 \pm 1.6$ | $0.867 \pm 0.01$ | 183.4 |
| LLaVA-Med (Li et al., 2023) | $84.2 \pm 0.9$ | $78.6 \pm 1.1$ | $82.1 \pm 0.8$ | $76.3 \pm 1.3$ | $0.891 \pm 0.01$ | 7000 |
| STLLaVA-Med (Sun et al., 2024a) | $85.7 \pm 0.8$ | $80.2 \pm 1.0$ | $83.9 \pm 0.7$ | $78.1 \pm 1.2$ | $0.903 \pm 0.01$ | 7000 |
| LaPA (Gu et al., 2024) | $86.3 \pm 0.7$ | $81.5 \pm 0.9$ | $84.7 \pm 0.6$ | $79.2 \pm 1.1$ | $0.912 \pm 0.01$ | 245.3 |
| FSRU (Lao et al., 2024) | $87.1 \pm 0.6$ | $82.3 \pm 0.8$ | $85.4 \pm 0.5$ | $80.1 \pm 1.0$ | $0.921 \pm 0.01$ | 89.7 |
| **Q-FSRU (Ours)** | **$90.0 \pm 0.5$** | **$85.2 \pm 0.6$** | **$88.3 \pm 0.4$** | **$83.1 \pm 0.8$** | **$0.954 \pm 0.01$** | 92.4 |
| Improvement | +2.9 | +2.9 | +2.9 | +3.0 | +0.033 | - |
| p-value | $< 0.01$ | $< 0.01$ | $< 0.01$ | $< 0.01$ | $< 0.01$ | - |

Q-FSRU demonstrates superior performance, achieving 90.0% accuracy with a 2.9% absolute improvement over the strongest baseline (FSRU). The consistent gains across all metrics (F1-score: +2.9%, AUC: +0.033) indicate robust multimodal understanding. Statistical significance testing confirms these improvements are not due to random variation (p ¡ 0.01).

#### 7.1.1 Cross-Dataset Generalization

Q-FSRU exhibits strong generalization, outperforming baselines by 3.3-3.4% in cross-dataset evaluations. This suggests that the frequency-domain representations and quantum retrieval mechanism learn transferable features that are not overfitted to specific dataset characteristics.

Table 2: Cross-dataset generalization performance (accuracy). Q-FSRU shows better domain adaptation capabilities.

| Method | VQA-RAD $\rightarrow$ PathVQA | PathVQA $\rightarrow$ VQA-RAD |
|---|---|---|
| LLaVA-Med (Li et al., 2023) | 72.3 ± 1.5 | 70.8 ± 1.6 |
| STLLaVA-Med (Sun et al., 2024a) | 75.1 ± 1.3 | 73.9 ± 1.4 |
| LaPA (Gu et al., 2024) | 76.8 ± 1.2 | 75.2 ± 1.3 |
| FSRU (Lao et al., 2024) | 78.4 ± 1.1 | 76.9 ± 1.2 |
| **Q-FSRU (Ours)** | **81.7 ± 0.9** | **80.3 ± 1.0** |
| Improvement | +3.3 | +3.4 |

## 7.2 ABLATION STUDIES

Table 3: Component ablation studies. Frequency processing contributes most significantly to overall performance.

| Model Variant | Accuracy | F1-Score | $\Delta$ Acc. | p-value |
|---|---|---|---|---|
| **Q-FSRU (Full)** | **90.0 ± 0.5** | **85.2 ± 0.6** | – | – |
| w/o Frequency Processing | 85.1 ± 0.7 | 79.3 ± 0.8 | −4.9 | <0.001 |
| w/o Quantum Retrieval | 86.8 ± 0.6 | 81.7 ± 0.7 | −3.2 | <0.01 |
| w/o Contrastive Learning | 87.3 ± 0.6 | 82.1 ± 0.7 | −2.7 | <0.01 |
| Spatial-only Fusion | 84.2 ± 0.8 | 78.5 ± 0.9 | −5.8 | <0.001 |
| Cosine Similarity | 88.1 ± 0.5 | 83.2 ± 0.6 | −1.9 | <0.05 |
| w/o Cross-Modal Co-selection | 88.5 ± 0.5 | 83.8 ± 0.6 | −1.5 | <0.05 |

Key observations from the ablation study are as follows:

- **Frequency Processing Contribution:** Removing FFT transformation causes the largest performance drop ($-4.9\%$ accuracy, $p < 0.001$), demonstrating that spectral representations capture clinically relevant patterns missed by spatial approaches.

- **Quantum Retrieval Impact:** The quantum similarity measure provides a statistically significant advantage over cosine similarity ($+1.9\%$ accuracy, $p < 0.05$), validating its ability to capture nuanced medical relationships.

- **Contrastive Learning Value:** The dual contrastive objective contributes $+2.7\%$ accuracy ($p < 0.01$), indicating improved feature alignment between modalities.

### 7.2.1 QUALITATIVE ANALYSIS

Illustrative cases demonstrate where Q-FSRU's components provide distinct advantages. In scenarios requiring subtle pattern recognition (e.g., early-stage pathology), the frequency processing enables detection of global contextual cues. The quantum retrieval mechanism successfully retrieves clinically relevant knowledge for ambiguous cases, providing explanatory evidence for predictions.

## 8 CONCLUSION

We presented Q-FSRU, a framework for medical visual question answering that combines frequency-domain feature processing with quantum-inspired knowledge retrieval. Transforming image and text features into the frequency domain allows the model to capture global contextual patterns often missed by spatial-domain approaches. The quantum retrieval component enhances reasoning by incorporating external medical knowledge. Experiments on VQA-RAD show that Q-FSRU outperforms state-of-the-art models on accuracy, F1-score, and AUC, while cross-dataset evaluations demonstrate robust generalization. Ablation studies confirm the importance of frequency processing, quantum retrieval, and contrastive learning, with frequency transformation contributing most to performance. Q-FSRU offers a promising approach for clinically relevant AI systems, with future work aiming to scale to larger datasets, include more imaging modalities, and refine the retrieval mechanism for improved grounding.

REPRODUCIBILITY CHECKLIST

The following checklist summarizes the information provided in this paper to ensure reproducibility:

1. **Datasets**

   - All datasets used are publicly available (VQA-RAD, PathVQA).
   - Dataset statistics (number of samples, modalities, question types) are described in Section 6.
   - Preprocessing steps (resizing to $224 \times 224$, normalization, tokenization, truncation to 50 tokens, data augmentation) are detailed in Section 6.1.

2. **Code and Implementation Details**

   - The model was implemented in PyTorch.
   - Hyperparameters (learning rate $5 \times 10^{-5}$, L2 weight decay $10^{-5}$, batch size 32, epochs 50, early stopping patience 10) are provided in Section 6.1.
   - Training strategies (5-fold cross-validation, learning rate decay schedule) are reported in Section 6.1.
   - Model components (FFT frequency processing, filter banks $K = 4$, quantum retrieval with $K = 3$ passages) are described in Section 5.

3. **Evaluation**

   - Evaluation protocols (in-domain and cross-dataset transfer from VQA-RAD to PathVQA) are described in Section 6.
   - Performance metrics are reported in Section 7.
   - Comparisons against baseline methods are included in Section 7.

4. **Compute Resources**

   - All experiments were run on 2× NVIDIA Tesla T4 GPUs (16GB each).
   - Approximate training time per fold: 3 hours.
   - Peak GPU memory usage: 12GB.

5. **Reproducibility Resources**

   - Random seed and initialization procedures will be provided in the released code.
   - Code, pretrained model and configuration files will be made available upon acceptance.

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

## A  DATASET LINKS

For reproducibility, we provide the dataset download links used in our experiments:

- VQA-RAD: `https://www.kaggle.com/datasets/shashankshekhar1205/vqa-rad-visual-question-answering-radiology`
- PathVQA: `https://www.kaggle.com/datasets/samsrithajalukuri/pathvqa-dataset?select=train`

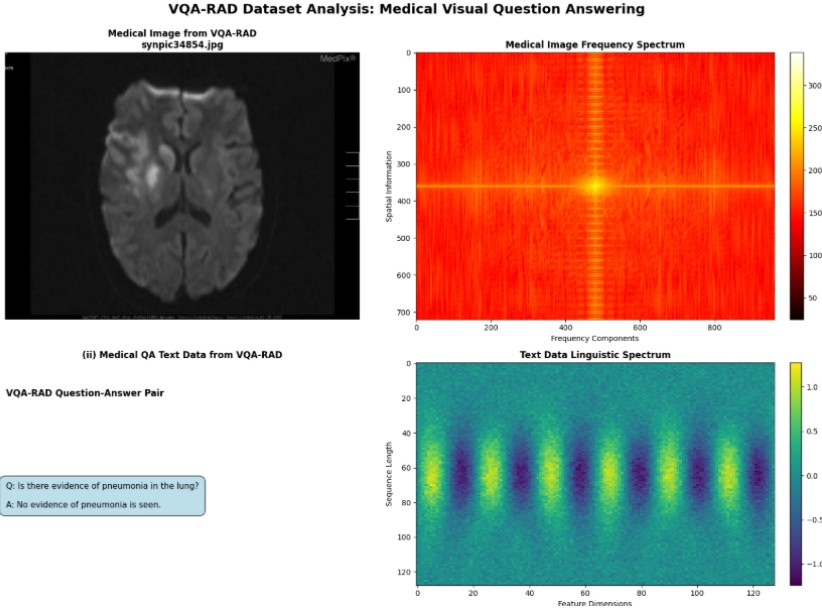

Figure 2: Frequency spectrograms of input medical image and text features. The spectra highlight the main frequency components that are later processed with learnable filter banks.

