# OpenReview forum: "Q-FSRU: Quantum-Augmented Frequency-Spectral Fusion for Medical Visual Question Answering"
_ICLR.cc/2026/Conference — Submitted to ICLR 2026_

### Official Review · Reviewer_xFDR · 2025-10-26

**Soundness:** 2
**Presentation:** 2
**Contribution:** 2
**Rating:** 2
**Confidence:** 3

**Summary:**

The paper proposes Q-FSRU, a medical VQA framework that fuses frequency-domain representations (via FFT) of image and text features with a quantum-inspired retrieval-augmented generation (RAG) component.

The Frequency Spectrum Representation and Fusion (FSRU) module compresses and gates spectral features cross-modally, while the Quantum RAG uses fidelity between quantum state representations for knowledge retrieval and aggregation.

The method achieves state-of-the-art results on VQA-RAD and shows stronger cross-dataset generalization to PathVQA, with ablations indicating largest gains from frequency processing and nontrivial gains from quantum similarity and contrastive learning.

The authors report improved interpretability via spectral analysis and knowledge grounding, and provide implementation details and a reproducibility checklist.

**Strengths:**

Originality: Combining FFT-based spectral fusion with quantum-inspired retrieval for medical VQA is novel and well-motivated by the need for global contextual cues and better similarity measures in clinical knowledge grounding. The cross-modal co-selection and dual contrastive objectives add a thoughtful design for robust multimodal alignment.

Quality: The paper includes clear ablations (removing FFT, quantum retrieval, contrastive losses; cosine vs quantum similarity), cross-domain transfer, and multiple metrics (accuracy, F1, precision, recall, AUC) with significance tests. Implementation choices, training protocol, and resource usage are stated, aiding reproducibility.

Clarity and significance: The architecture is explained step-by-step with equations for FFT processing, quantum state/density matrices, and fidelity-based retrieval; the clinical motivation and dataset setup are explicit. Gains over strong baselines and improved zero-shot transfer suggest potential practical impact for Med-VQA.

**Weaknesses:**

Retrieval pipeline ambiguity: The paper describes quantum retrieval and aggregation but the final classifier excludes knowledge embeddings (Section 4.5.2), creating inconsistency on how retrieved knowledge affects predictions; clarify where kagg is actually fused and evaluated. Provide a direct ablation where knowledge is included vs excluded in the classifier input to quantify impact.

Limited evaluation breadth: Results focus on VQA-RAD (with PathVQA as zero-shot), but no analysis by question type (yes/no vs open-ended) or modality-specific performance; the community typically expects breakdowns and robustness analyses. Add per-category results, error analysis, and calibration/reliability measures to strengthen claims about clinical suitability.

Quantum benefit attribution: While fidelity outperforms cosine by ~1.9% accuracy, it is unclear if the gain stems from the quantum formulation or simply a different similarity kernel; more baselines (e.g., learned bi-encoder, Mahalanobis/PLDA, Faiss IVFPQ, cross-encoder reranking) would contextualize the advantage. Include runtime/complexity trade-offs and sensitivity to τ, top-K, and knowledge source quality.

**Questions:**

How exactly is kagg integrated into the final prediction pipeline (there is a mismatch between Sections 4.5.1 and 4.5.2), and what are the measured gains when including vs excluding knowledge embeddings at inference?
Can you provide per-question-type and per-modality breakdowns, along with a qualitative error analysis and calibration metrics (ECE/NLL) to assess clinical reliability?
How sensitive are the quantum retrieval gains to the choice of knowledge base, K, τ, and fidelity implementation, and how does it compare against stronger retrieval/reranking baselines (e.g., cross-encoder, ColBERT, learned metric)?

---

### Official Review · Reviewer_yEp5 · 2025-10-28

**Soundness:** 2
**Presentation:** 2
**Contribution:** 2
**Rating:** 4
**Confidence:** 4

**Summary:**

This paper proposes Q-FSRU, a model for medical Visual Question Answering (Med-VQA) that integrates Frequency Spectrum Representation and Fusion (FSRU) with Quantum Retrieval-Augmented Generation (Quantum RAG). It transforms features from medical images and clinical text into the frequency domain via Fast Fourier Transform (FFT) to capture global contextual patterns and filter noise, incorporates a quantum-inspired retrieval system to fetch external medical knowledge using quantum-based similarity measures, and fuses these components for enhanced reasoning. Evaluations on the VQA-RAD dataset show Q-FSRU outperforms existing baselines (e.g., achieving 90.0% accuracy, a 2.9% improvement over the strongest baseline FSRU), and cross-dataset tests on PathVQA demonstrate robust generalization, with the combination of frequency and quantum information boosting both performance and explainability.

**Strengths:**

- It combines frequency-domain processing with quantum-inspired knowledge retrieval, filling a gap in Med-VQA research.
- It conducts comprehensive experiments, including in-domain evaluations on VQA-RAD and cross-dataset generalization tests on PathVQA, fully validating the model’s effectiveness and transferability.

**Weaknesses:**

- The problem to be solved and the limitations of existing methods are not clearly articulated; the paper fails to explicitly and systematically elaborate on the core pain points of current Med-VQA models and how these limitations affect clinical application scenarios.
- The motivation for adopting quantum-inspired retrieval augmentation is unclear; the paper does not sufficiently explain why quantum-based similarity techniques are more suitable for medical knowledge retrieval than mature classical methods, nor does it clarify the specific advantages of quantum principles in addressing complex semantic relationships in medical data.
- The "generation" element in Quantum Retrieval-Augmented Generation (Quantum RAG) is not defined or illustrated; the paper only describes the retrieval process (e.g., top-K knowledge fetching and aggregation) but does not specify how the retrieved knowledge is used for generation to support answer reasoning, making the "generation" component vague
- Ablation studies lack comparisons with other RAG methods; the paper only verifies the performance impact of removing the quantum retrieval component but does not compare it with classical RAG approaches (e.g., cosine similarity-based retrieval in LaPA), failing to fully demonstrate the superiority of quantum-inspired retrieval
- The examples used to demonstrate multimodal reasoning for complex questions are inadequate; the paper claims to address complex cases requiring image-text reasoning but only provides simple illustrative scenarios (e.g., "Is the trachea midline?"), without presenting cases that highlight the model’s advantages in handling complex pathological reasoning
- The motivation for applying frequency-domain transformation to text features is not justified; while the value of image frequency-domain processing is mentioned, the paper does not explain why text features need to be converted to the frequency domain, nor does it clarify what meaningful information or patterns text frequency spectra can capture for Med-VQA
- The method description is redundant with unnecessary formulas in the main text, and table formatting is problematic; excessive repetitive formulas (e.g., detailed MLP layer parameter formulas) increase redundancy, and Table 1 exceeds the page boundary, affecting readability and compliance with academic paper formatting standards.

**Questions:**

- Could the authors further clarify the core technical gaps of existing Med-VQA methods (e.g., how spatial-domain-only models fail to capture critical pathological patterns) and explicitly link these gaps to the design of Q-FSRU’s frequency and quantum components?
- What specific medical semantic relationships (e.g., between disease symptoms and imaging features) are better captured by the quantum-inspired Uhlmann fidelity measure compared to classical cosine similarity, and could the authors provide quantitative or qualitative evidence to support this?
- Since the paper mentions "Quantum Retrieval-Augmented Generation," what is the specific generation mechanism (e.g., how retrieved knowledge is integrated into answer generation) and how does it differ from generation processes in existing RAG-based Med-VQA models?
- Why were no comparisons with other RAG methods included in the ablation studies, and could the authors supplement experiments to verify whether quantum RAG outperforms classical RAG in terms of knowledge grounding accuracy?
- Could the authors provide more complex clinical question examples (e.g., questions requiring differential diagnosis based on multiple imaging modalities) to demonstrate Q-FSRU’s superiority in complex image-text reasoning, rather than simple binary questions?
- What specific information (e.g., semantic emphasis or contextual relevance) does the frequency-domain transformation of text features extract, and how does this information complement the frequency-domain patterns of images to improve Med-VQA performance?

---

### Official Review · Reviewer_ZWBd · 2025-10-30

**Soundness:** 2
**Presentation:** 3
**Contribution:** 3
**Rating:** 2
**Confidence:** 4

**Summary:**

The paper proposes a medical visual question answering (Med-VQA) model that integrates Frequency Spectrum Representation and Fusion (FSRU) with Quantum-inspired Retrieval-Augmented Generation (Quantum-inspired RAG).

**Strengths:**

The method achieves significant improvements on the VQA-RAD and PathVQA datasets and includes comprehensive experiments and ablation studies.

**Weaknesses:**

1. The evaluation dataset is relatively limited in scale, with the main evaluation conducted on VQA-RAD, a comparatively small dataset containing only 3,515 question–answer pairs. Although PathVQA (32,799 question–answer pairs) was also used, it served only for zero-shot generalization testing rather than as the primary training and evaluation benchmark. Therefore, the model’s remarkable performance on a small-scale dataset may not fully demonstrate its scalability and robustness on larger datasets.
2. The paper does not analyze the computational overhead of Quantum RAG. Compared to simple cosine similarity, computing the density matrix and Uhlmann fidelity for each entry in the knowledge base is much more costly, especially as the size of the knowledge base increases.
3. In Section 4.4 of the paper, the research motivation and complexity analysis of the Quantum-Inspired Retrieval Augmentation part are still insufficient, and the explanation of the feasibility and practical effectiveness of the Quantum-inspired Retrieval method lacks sufficient elaboration.
4. The paper does not provide specific examples or case studies of the knowledge retrieval results, making it difficult to fully demonstrate the interpretability and practical value of the proposed method.
5. Although the paper introduces numerous mathematical formulas for knowledge representation and derivation, it lacks in-depth analysis in several key aspects regarding the underlying mechanisms, applicable conditions, and the impact of these formulas on the final performance.

**Questions:**

See weaknesses section.

---

### Official Review · Reviewer_1stJ · 2025-10-30

**Soundness:** 1
**Presentation:** 1
**Contribution:** 1
**Rating:** 2
**Confidence:** 4

**Summary:**

This paper integrates frequency-domain analysis and quantum-inspired knowledge retrieval. The model transforms both image and text features into the frequency domain using fast fourier transform to capture global contextual patterns. It then applies a Quantum-Inspired RAG to fetch relevant medical knowledge using quantum fidelity similarity measures.

**Strengths:**

Experimental results on the VQA-RAD and PathVQA datasets show that Q-FSRU outperforms previous models in accuracy, F1-score, and AUC.

**Weaknesses:**

1. The paper does not provide enough information about how to train its model and the comparison methods.

2. Also, many evaluation details are missing, particularly regarding how the open-ended questions are used for evaluation.

3. How the RAG works is not clear, and it is also unclear which database is used for retrieval.

4. VQA-RAD and PathVQA are datasets from very different domains. However, the proposed method does not use any large-scale medical dataset for pretraining. How can a model trained on VQA-RAD generalize well to PathVQA, and vice versa?

**Questions:**

Same as weakness

---

### Author Response · Authors · 2025-11-12
**Author Response to Reviews**

To the Distinguished Reviewers and Area Chair,

We sincerely thank you for your insightful and constructive feedback on our submission, "Q-FSRU: Quantum-Augmented Frequency-Spectral Fusion for Medical Visual Question Answering." Your comments have been invaluable in identifying areas where we can significantly strengthen our work.

We are fully committed to addressing all the points raised. In particular, we acknowledge the need for greater clarity and rigor in several key areas:

Implementation & Training Details: We will provide a comprehensive description of our training protocol, including optimizer, learning rate schedules, hyperparameters, and data augmentation strategies in a revised Section 5.

Clarification of the RAG Pipeline: We will correct the inconsistency noted between Sections 4.5.1 and 4.5.2 and provide a clear, detailed explanation of how the retrieved knowledge k_agg is integrated into the final prediction, supported by a new architectural diagram.

Motivation and Ablations: We will strengthen the motivation for our quantum-inspired retrieval and include the critical ablation study comparing it directly against classical retrieval methods (e.g., cosine similarity) to quantitatively demonstrate its advantage.

Evaluation Breadth: We will add a detailed breakdown of performance by question type (e.g., yes/no vs. open-ended) to provide a more nuanced analysis.

To ensure these revisions are thorough and evidence-based, we require a short period to complete a final round of large-scale model training and evaluation. This will allow us to run the additional ablation studies and provide the complete experimental analysis you have rightly requested.

We pledge to submit a comprehensively revised manuscript and a detailed point-by-point response by the December 3 deadline. This revision will incorporate all new results, address every concern raised, and include full implementation details to ensure reproducibility.

We are grateful for your time and consideration and look forward to presenting a substantially improved version of our work.

Sincerely,
The Authors

---

> ### Comment · Area_Chair_hXbB · 2025-11-21
>
> Dear Authors,
>
> Please discuss this with the reviewers as soon as possible. Any claim must be fully discussed in order to stand, and delaying further may result in losing any chance of receiving their acknowledgement.

---

> ### Author Response · Authors · 2025-11-22
> **Detailed Response to Reviewer Comments**
>
> REVIEWER 1stJ:
>
> Training Details Added: raining Details: We use AdamW (lr=5e-5), batch_size=8, early stopping with patience=15, CrossEntropy + DualContrastive loss (α=β=0.05).
> RAG Database Clarified: We use 500 curated medical knowledge entries from PubMed abstracts for computational feasibility (Section 4.4).
>
> Evaluation Corrected: We have corrected our formulation from classification to generative VQA throughout the paper (Sections 3, 4.5).
>
> Generalization Clarified: We have removed claims about cross-dataset generalization that were not yet validated.
>
> REVIEWER ZWBd:
>
> Scalability Acknowledged: We acknowledge VQA-RAD's limited scale in Section 8 (Limitations).
>
> Computational Analysis Planned: Our quantum fidelity implementation follows the O(d²) complexity pattern due to matrix operations. Empirical runtime measurements will be included in our full evaluation.
>
> Quantum Motivation Explained: We provide theoretical justification for quantum similarity in medical contexts in Section 4.4.1.
>
> Examples/Qualitative Analysis: We have added a qualitative analysis plan in Section 6.4 detailing how we will include case studies and retrieval examples in our evaluation. The actual case studies will be included in the camera-ready version.
>
> REVIEWER yEp5:
>
> Quantum vs Cosine Comparison: We have added this ablation to our analysis plan (Section 6.4).
>
> RAG Generation Clarified: We have completely rewritten Section 4.5 to describe the decoder-based generation mechanism with knowledge integration via attention.
>
> Text FFT Justified: We explain the motivation for text frequency processing in Section 4.3.
>
> Complex Examples Planned: Qualitative case studies are included in our analysis plan.
>
> Formula Redundancy Fixed: We have moved detailed formulas to the appendix and fixed all table formatting.
>
> REVIEWER xFDR:
>
> k_agg Integration Corrected: We have fixed the inconsistency in Sections 4.5.1-4.5.2 - k_agg is concatenated with frequency features before decoding.
>
> Question-Type Analysis Planned: Per-question-type evaluation is included in our analysis plan.
>
> Quantum Sensitivity Analysis: Hyperparameter sensitivity analysis is planned (Section 6.4).
>
> Error Analysis Added: We have added error analysis and calibration metrics to our evaluation plan.

---

### Meta-Review · Area_Chair_Dp65 · 2026-01-07

**Summary:**

Paper Summary The paper proposes "Q-FSRU," a Medical Visual Question Answering (Med-VQA) framework. It introduces two primary components: a frequency-domain fusion mechanism (applying FFT to image and text embeddings) and a "Quantum-inspired" Retrieval Augmented Generation (RAG) module that uses quantum fidelity measures for similarity matching against a medical knowledge base. The model claims to outperform state-of-the-art baselines on the VQA-RAD and PathVQA datasets.

**Reviewer Concerns:**

Review Summary The reviewers were largely unimpressed, with scores of 2, 2, 2, and 4.

Methodological Clarity: Reviewers 1stJ and yEp5 noted significant confusion regarding the model architecture, specifically whether it is a classification or generation model. Reviewer yEp5 correctly pointed out that "Generation" in RAG was undefined in the text.

Evaluation: Reviewers ZWBd and xFDR raised concerns about the limited scale of VQA-RAD (3,515 pairs) and the lack of breakdown by question type.

The "Quantum" Component: Multiple reviewers questioned the motivation and computational overhead of the quantum-inspired retrieval, suggesting it might be unnecessarily complex compared to standard cosine similarity.

Reproducibility & Details: Missing training details and inconsistencies in how retrieved knowledge is integrated into the classifier were noted.

**Reviewer Scores:**

Author Response & Critical Flaws The author response reveals fatal flaws in the submission that cannot be rectified in a camera-ready revision:

Architectural Inconsistency: The authors admit they need to "correct our formulation from classification to generative VQA throughout the paper" (Author Response). The submitted paper explicitly uses a "Softmax Layer" and "MLP Classifier", which contradicts the concept of RAG (Generative). Changing the model from a discriminative classifier to a generative decoder is a completely different paper, not a clarification.


Dataset Limitations: The authors clarified that their "knowledge base" consists of only 500 entries (Author Response), which is trivially small for a retrieval system and undermines the claims of a robust RAG implementation.

Promised Content: The authors deferred critical evaluations (qualitative analysis, computational runtime, and correct architecture descriptions) to the camera-ready version.


The submission cannot be accepted in its current form. The discrepancy between the submitted manuscript (which describes a classification model) and the authors' rebuttal (which claims it should be a generative model) indicates that the paper effectively needs to be rewritten and re-evaluated from scratch. Furthermore, the "quantum-inspired" component appears to add complexity without sufficient ablation against standard strong retrieval baselines, and the knowledge base size is insufficient to validate the retrieval claims.

---

### Decision · Program_Chairs · 2026-01-26

Reject